# Selective Membrane Sensor for Aluminum Determination in Food Products, Real Samples and Standard Alloys

**DOI:** 10.3390/membranes11070504

**Published:** 2021-06-30

**Authors:** Sabry Khalil, Ashraf Y. Elnaggar

**Affiliations:** Department of Food Nutrition Science, College of Science, Taif University, P.O. Box 11099, Taif 21944, Saudi Arabia; aynaggar@Tu.edu.sa

**Keywords:** ion-associate Tetraaza Tetraene complexes, membrane sensor, aluminum estimation, food, real samples, alloys analysis

## Abstract

The study involves the fabrication of an aluminum liquid membrane sensor based on the association of aluminum ions with the cited reagent 2,9-dimethyl-4,11-diphenyl -1,5,8,12-tetraaza cyclote tradeca-1,4,8,11-tetraene [DDTCT]. The characteristics slope (58 mV), rapid and linear response for aluminum ion was displayed by the proposed sensor within the concentration range 2.5 × 10^−7^–1.5 × 10^−1^ M, the detection limit (1.6 × 10^−7^) M, the selectivity behavior toward some metal cations, the response time 10 s), lifetime (150 days), the effect of pH on the suggested electrode potential and the requisite analytical validations were examined. The suitable pH range was (5.0–8.0), in this range the proposed electrode response is independent of pH. The suggested electrode was applied to detect the aluminum ions concentration in food products, real samples and standard alloys. The resulting data by the suggested electrode were statistically analyzed, and compared with the previously reported aluminum ion-selective electrodes in the literature.

## 1. Introduction

Aluminum can be found in combination with other elements, and cannot be excreted by patients with kidney failure diseases. In the dialysis fluid or the long-time medical treatment toxicity associated with exposure to aluminum is now recognized [1]. Care is taken to check blood levels of aluminum in patients with kidney failure. Aluminum can cause toxicity in humans, and is also a valuable factor in pathological diseases [2,3,4,5,6]. Thus, it is very significant to estimate aluminum in pharmaceuticals, real samples, alloys and food stuff.

Utilities of electrode sensors have developed to use in different ways [7,8,9]. Many techniques, like, atomic absorption spectrometry; AAS [10,11,12,13,14,15,16], spectro-fluorometry [17,18], chromatography [19,20,21,22], a laser diode atomic absorption spectroscopy [23], inductively coupled plasma-atomic emission spectrometry; ICP-AES [24,25,26], inductively coupled plasma-mass spectra; ICP/MS [27,28,29,30], spectrophotometry based on use of various reagents [31,32,33,34,35,36,37,38,39,40,41], ion-selective electrode techniques [42,43,44,45,46,47,48,49,50,51,52] and other recently different methods [53,54,55,56,57,58,59,60,61,62,63] have been listed for the estimation of aluminum. Although those reported methods are highly sensitive for the determination of aluminum ions, some difficult complications were produced in their applications. Voltammetry is a very ponderous procedure in spite of it is economic. Potentiometric estimation based on the ion-selective membrane electrode is very simple and introduced several good characteristics, like wide linearity in the concentration range, easy sampling, rapid response, simple equipment, highly selective with a very low value of detection limit, carried out in turbid, colored, and/or viscous solutions and less cost. Anyway, many of the previously reported electrodes suffering from interfering in calcium ions and have a limited range of concentration. 

New reagents from heterodiazo dyes which form very stable and strong ion-associates with some active ions of some metals were designed to determine them in pharmaceutical and food products by new, very sensitive, and selective spectrophotometric detection [64,65]. 

The reagent 2,9-dimethyl-4,11-diphenyl-1,5,8,12-tetra azacyclotetra deca-1,4,8,11-tetraene [DDTCT] contains the functional groups tetradeca, and tetraene have attracted considerable attention as a synthetic ionophore, as this armed macrocycle, as a result of its structural characteristics, binds metal cations with different strengths, and thus, can be convenient as electro active material to apply in selective sensor. This compound has been utilized for the estimation of a numerous metal cations [66]. Therefore, we decided to use it in the designation of a new aluminum electrode.

The present work describes the construction and evaluation of aluminum(III) membrane sensor. The active constituents in a polyvinyl chloride; (PVC) matrix selective sensors are the Al^3+^ with the cited reagent [DDTCT] ion associate complex. The characteristics slope, linear and rapid response for aluminum ion was presented by the developed ion-selective electrode within the concentration range 2.5 × 10^−7^–1.5 × 10^−1^ M, the limit of detection limit (1.6 × 10^−7^) M, the selectivity behavior toward some metal cations, the lifetime (150 days), the response time (10 s), the influence of pH on the sensor potential and the interesting analytical parameters were examined. The suitable pH range was (5.0–8.0); the response of the proposed electrode is not affected by pH change in this range. The sensor is successfully utilized for the estimation of the concentration of aluminum (III) ions in food products, real samples and standard alloys.

The produced data by the suggested electrode were analyzed, and compared with those of various published aluminum ion-selective electrodes.

## 2. Chemicals and Methods

### 2.1. Product Samples, Chemicals and Reagents

Aluminum, cadmium, sodium, zinc, nickel, cobalt, and calcium chlorides, ammonium and sodium hydroxide, hydrogen peroxide and polyvinyl chloride; PVC were Aldrich products. Hydrochloric, hydrofluoric and sulfuric acids, tetrahydrofuran, TBP(tributylphosphate), acetate buffer solution and methanol from Merck Germany. Food products containing aluminum (Bread, Flour, Rice, Tea-Leaves, Tomato Sauce and Chocolate) were purchased from the local stores and markets in Saudi Arabia and Egypt, real samples (Granite, Basalt, Rhyolite), and standard alloys (Copper-based alloy NBS164 and Zinc-based alloy NIST94C). 

### 2.2. Preparation of Stock Solutions

Chloride stock solutions of aluminum, zinc, nickel, cadmium, sodium, cobalt, and calcium of 0.1 molar solution were weighed and dissolving the calculated quantities of each one in distilled H_2_O. 10^−7^–10^−1^ molar solutions were conditioned by dilution.

Standard solutions of aluminum chloride utilized in the estimation of aluminum in, food products, real and standard alloys samples were weighed, a calculated quantity of each sample was prepared in 0.01 molar sodium chloride and conditioned by dilution.

### 2.3. Sampling for Aluminum Ions Determination

The prerequisite solutions for potentiometric measurements were obtained as follows: a content of food products (Bread, Flour, Rice, Tea-Leaves, Tomato Sauce, and Chocolate) were selected for analysis. For the analysis of Al^3+^ ions in bread and chocolate, samples were cut, washed, then heated at 120 °C for 2 h. Weight strictly 10 grams, transferred (460 and 400 mg of them, respectively) into a crucible, heated at 600 °C for 4 h. for ashing, after entiring the ashing the samples were cooled to the ambient temperature, 5 mL of diluted HCl were added for dissolving the resulting residues, put into a 50 mL calibrated marked flask and diluted with distilled H_2_O. For flour and rice, 10-grams sample was accurately weighed, transferred (750 and 1500 mg, respectively) to a quartz crucible. 10 mL concentrated nitric acid was added, evaporated to dryness. By heating concentrated H_2_O_2_ was added gradually till a clear solution is obtained, then evaporated. To eliminate H_2_O_2_ distilled water was added, and heating constantly. The produced yield was cooled, diluted with distilled water. For tea- leaves and tomato sauce analysis, five-gram samples were weighed accurately, (550 and 175 mg of them, respectively) was taken and follow the same procedure as mentioned above. Taking 10-mL of each aqueous solution for Al^3+^ ions estimation adjusted at the ideal circumstances of the proposed mode and the providing results are tabulated in Section 3.6.

For Al^3+^ ion determination in real samples, 0.5 g of the mashed selected sample of some rocks (Basalt, Rhyolite and Granite) was dissolved in 10 mL of 20 molars hydrofluoric acid solution under gentle heating condition till dryness. (500 mg of each of them) was transferred into a conical flask, dissolving the residue in 5 mL of 4 molars sulfuric acid, diluted to 25 mL, then diluted 250 times with distilled water. Adjusting the pH at 5 using an acetate buffer solution.

In the case of the analysis of standard alloys, weighed accurately the percent content alloy, transferred (250 and 300 mg of copper and zinc-based alloy, respectively) into a conical flask, and dissolved completely in 40 mL of hydrochloric acid with heating, adding 3 mL of 30% H_2_O_2_. The mixture was cooled and filtered and diluted to 500 mL with distilled H_2_O. For each of the alloy solutions, 10 mL were taken for the estimation of Al^3+^ ions applying the proposed method at the ideal circumstances. The resulting data are reported in Section 3.6.

### 2.4. Membrane Electrode Fabrication

The fabricated electrode was displayed as published before [67]. It contains a column sensor of Teflon commutable and a form full of a liquid phase sensor “+ Ag/Ag Cl” (an internal reference electrode). 

The complex, plasticizer and the polyvinylchloride; PVC were grounded, adding tetrahydro furan as a flown solvent. A convenient diameter disk was cut and glued to the flat end of polyvinyl chloride; PVC tubing with Tetrahydrofuran; THF. The form of the sensor was wind with 0.001 molar specific solution of aluminum membrane electrode. The developed electrode was conditioned by soaking for 24 h in 0.01 molar Al^3+^ solution and kept for a remnant duration in a similar solution. 

### 2.5. Working Constituent of Liquid-Electrode Coat

The cited reagent 2,9-dimethyl-4,11-diphenyl-1,5,8,12-tetraazacyclo tetradeca-1,4,8,11-tetraene[DDTCT] is a white powder active membrane component. It dissolved completely in diluent trahydrofuran (15 mL) and made up with methanol to 100 mL, concentration = 5 × 10^−4^ molar at pH 5.0–8.0. The cited reagent was conditioned as described before [66].

### 2.6. Preparation of the Potential Layer

A precise weight 0.02 g active constituent [Al(DDTCT)] mixed with 0.35 g PVC, and 0.63 g TBP were grounded to provide the suggested electrode’s coat. A Teflon electrode with a reference of Ag/AgCl was completely wined with the recently conditioned mixture, then transferring to gel by heating at a suitable temperature of 375 K for 20 min. After cooling, the suggested sensor was conditioned for two hrs., into a 10^−3^ molar solution aluminum ion.

### 2.7. EMF Measurements

An Orion 90-02 reference electrode was applied with a mechanical stirrer to provide a veracity of 0.1 mV at the laboratory conditions to measure the EMF of the aluminum electrode regulation. An Orion 90-00-01 solution including 0.05 M sodium chloride, 1.5 M potassium nitrate, 0.55 M potassium chloride, and 40% formaldehyde one ml was utilized to complete filling the stabilized bridge of the reference electrode. 

## 3. Results

The analytical validation of our designed aluminum sensor were investigated to estimate its prominence in practical utilities. The selectivity characteristics, detection limit, the observed slope, the restraint time, and dependence of pH on the electrode potential response were also examined.

### 3.1. The Calibration Curve of the Suggested Electrode

Figure 1 presented the developed aluminum electrode’s calibration curve detected in aluminum and its interfering ions of 10^−7^–10^−1^ molar solutions.

The aluminum sensor’s characteristics slope is 20 mV, the limit of detection is 1.6 × 10^−7^ molar and the measuring range is 2.5 × 10^−7^–1.5 × 10^−1^ molar. Table 1 summarised the analytical characteristic parameters of the constructed aluminum sensor.

### 3.2. Selectivity Behavior

The selectivity behavior of the Al^3+^ sensor with various interfering ions such as Cd, Ni, Co, Ca, and Cu with the same concentration of aluminum 10^−3^ molar were examined applying the modes of the separate solution or the MP, (matched potential) listed previously [67] applying the equations:(1)$$\log\;K^{\mathrm{pot}}{\;}_{\mathrm{ij}}=E_{2}\;-E_{1}/S-(Z_{i}/Zj_{\;}-1)\;\log\;a_{i},\;K^{\mathrm{pot}}\;\mathrm{Al}/M=\frac{\mathrm{ai}}{\mathrm{ai}\frac{\mathrm{zi}}{\mathrm{zj}}}$$

By using the separate solution method, at the EMF value of Al^3+^ ions with the concentration 0.001 M and, the potential –160 mV. For the matched potential mode, the equation is: $$K^{\mathrm{pot}}\;\mathrm{Al}/M=\frac{\mathrm{ai}}{\mathrm{ai}\frac{\mathrm{zi}}{\mathrm{zj}}}$$

The provided results are tabulated in Table 2.

### 3.3. Response Time of the Suggested Aluminum Electrode

For analytical validations, the restraint time of the fabricated selective developed electrode is very important. For dilution adding water (1:1) after injecting the standard concentrated solution, Solutions used for the determination of the restraint time of the suggested tested electrode have the following conditions: v_1_:v_2_ = 1: 20,c_1_:c_2_ = 1:100, where v_1_ is the quantitative amount of the sample, and v_2_ is the standard quantitative amount, c_1_ is the sample concentration, c_2_, the concentration of the standard. The obtained data are presented in Figure 2. After 12 seconds of adding aluminum the response of the sensor is reproducible. At the instant of injection of the concentrated sample the timer is started, the quick and stable reading of potential reflecting the time required for ending the titration. The ion-selective electrode scopes its balance in a shortened response time (12 s) over the whole linear of the applicable concentration range as shown from Figure 2.

### 3.4. pH Influence on the Proposed Electrode Potential

The influence of the sensor potential on the pH was investigated by tracking the potential measurements with respect to the chemical property of aluminum salts. Stepwise of sodium hydroxide or hydrochloric acid or were provided to the investigated 0.001 M aluminum ions sample. the pH was reported after each increment, the ratio of the electromotive force; EMF of the aluminum membrane system/reference electrode was measured after the sensor’s restraint was attained. The influence of pH on the EMF is presented in Figure 3. Below and above this pH range (5.0–8.0), at higher pH values, the potential decreases (−187 at pH 9, −192 at pH 9.5, and −198 at pH 10) may be due to the hydrolysis of Al^3+^ ions or the complex formation is not completing. At lower pH values, potential increases (−125 at pH 2.4, −134 at pH 2.7, and −140 at pH 3) attributed to the membrane responses to hydronium. and/ or Al(III) ions.

### 3.5. Duration time of the Aluminum Sensor

The duration time of the investigated aluminum sensor under studying was tested by measuring the specific slopes of the membrane electrode kept in 4 °C. The regular investigations were completed once a week for 7 months, in recently conditioned solutions in a consistent mode. Stabilized and repeatable signs were obtained through 5 months. It was noticed that a trivial decrease in the sensor slope by 1.0 mV decade^−1^ from 58.00–57.00 mV/decade and an increment in the limit of detection value. By the end of the time period, the slope of the electrode decreased gradually, whereas the detection limit is increased to become (from 52.34 to 48.16 mV per decade and 3.2 × 10^−6^ to 4.5 × 10^−5^ M, respectively) were observed. This probably arises from the leaching of the electrode constituents. Thus, the long duration time of the sensor is about 5 months, according to the subsistence of the provided results.

### 3.6. Determination of Aluminum in Foodstuff, Real Samples, and Standard Alloys

The dissection of Al^3+^ ions in foodstuff, real samples, and standard alloys was examined applying the suggested sensor to study its practical utility. The processes of standard additions and that of the calibration curve were applied, the detected data and their analyzed validation are presented in Table 3.

## 4. Discussion

The characteristic Nernstian slope is a quite paramount parameter to evaluate eclectic sensors that are specifically applied in the analysis. The optimum value of the Nernstian slope is 59.1/n (mV/decade), where (n) is the valency [68]. This indicates that for Al^3+^ electrode with the n value = 3 is 19.7 mV/decade. The value of the specific Nernstian parameter this work was 19.33 mV/decade, indicates an increment in the molarity of 10^−1^ M solutions under-test, the potential alteration of 19.33 mV/decade. This proves that the Al^3+^ developed electrode is still workable for the dissection of Al^3+^ application, because the acceptable applicable value of the specific Nernstian slope is 19.7 mV. 

In this work, various cations were tested as foreign interfering ions. The values of selectivity coefficients presented in Table 2 indicated that the developed aluminum ion electrode is strongly eclectic to aluminum(III) safely in the existence of Co, Ca, Cu, Cd, and Ni. As shown from the obtained results, none of the investigated interfering cations had a noticed effect on the reaction of the potentiometric measurements of the membrane sensor towards aluminum ions. Obviously, for all applied different cations, the selectivity coefficients values are smaller, providing they would not safely hinder the work of the suggested Al^3+^ electrode. The surprise of the noticed great selectivity of the sensor towards Al^3+^ ions in the existence of other tested cations is attributed to the greatest ability of the carrier molecules for aluminum ions.

Also, as summarized in (Table 3) that the calibration curve and the standard additions methods were utilized. The analysis of the resulting data showed that the calibration curve method is more preferable in the estimation of Al^3+^ while the method of standard additions is less preferred, the noticed error is about 1.91 %, and 1.0 % in the two ways, respectively, which is attributed to the repeatability, veracity, and reproducibility of the style.

The results provided by the fabricated Al^3+^ electrode were analyzed, and compared with the previously published ion-selective sensor. The data regarded in Table 4 compare between some of the important parameters of the quantitative estimation of Al^3+^ ions applying various selective sensors published before to indicate that the developed sensor performs acceptable good finding and be utilized for Al^3+^ ions estimation in foodstuff, real samples and standard alloys. As presented in Table 4 the developed sensor exhibits a comparable linear concentration range (2.5 × 10^−7^–4.5 × 10^−1^ M) which is more valuable than the other previously reported Al^3+^ selective sensors [2,7,42,43,44,45,46,47,48,49,50]. It has a longer life span (150 days ) in comparison to the previously reported sensors, those have low detection limits, the lowest one is that introduced in our work (1.6 × 10^−7^)even with the nearest linear concentration range. Further, the developed sensor has numerous advantages in comparison to others sensors, it is easier to construct, it is low cost. Thus, it can be reliable to state that our proposed electrode is acceptable to use with other sensors for Al^3+^ ions estimation. 

No detected interference was attained from the constituents that existed in the investigated samples. The curve of calibration presented an excellent response to the linearity range of concentrations. Several methods introduce valuable results compared to the real values and there is no noticeable variations were shown for either precision or accuracy.

## 5. Conclusions

The constructed aluminum selective sensor was developed. The proposed membrane sensor has many excellent analytical characteristics: relatively short response time, long duration-lifetime, and the Nernstian slope. Table 1 and Table 2 displayed the analytical characteristics of our investigated new sensor.

The constructed sensor was applied for Al^3+^ ions estimation in food stuffs, real samples, and standard alloys which utilized in common. The method of calibration curve and that of the standard additions were utilized. The analyzed results have confirmed that the calibration curve method is better and more preferable than the standard additions method in the aluminum estimation. However, the noticeable error is not bigger than 1% which is attributed to the repeatability and reproducibility of the applied method. The method described in Table 4 for the proposed sensor was sufficiently precise, and accurate in comparison with the other described that are utilized in common for aluminum ions estimation in foodstuff products, environmental real samples, and standard alloys.

The proposed ion-selective electrode was applied for Al^3+^ ion estimation in the selected samples of foodstuff, real, and standard alloy that utilized in common. The methods of standard additions and that of the calibration were utilized. The analyzed results showed that the calibration curve way is better in the deduction of foodstuff, real, and standard alloy samples than the way of standard additions which is less preferred. Therefore, due to the reproducibility and repeatability of the proposed method, the error is no bigger than 2%. 

Thus, the results exhibited excellent quality which is attributed to the good luck with the selection of samples applying the developed selective proposed sensor. The time consumed through the estimation is tested alone with no considerable influence on the veracity, reproducibility, and exactitude of the obtained data.

## Figures and Tables

**Figure 1 membranes-11-00504-f001:**
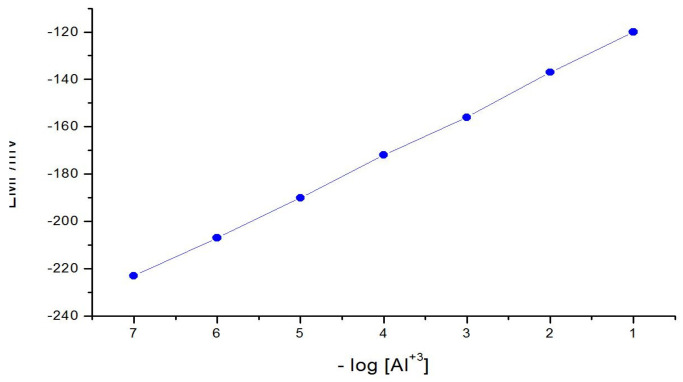
The curve of calibration of the developed Al(III) electrode over the concentration scope 10^−7^–10^−1^ M.

**Figure 2 membranes-11-00504-f002:**
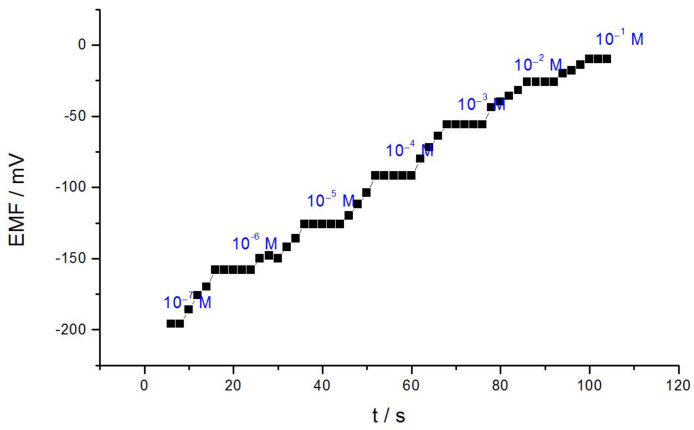
The time of response of [10^−3^ M] concentration of proposed aluminum electrode.

**Figure 3 membranes-11-00504-f003:**
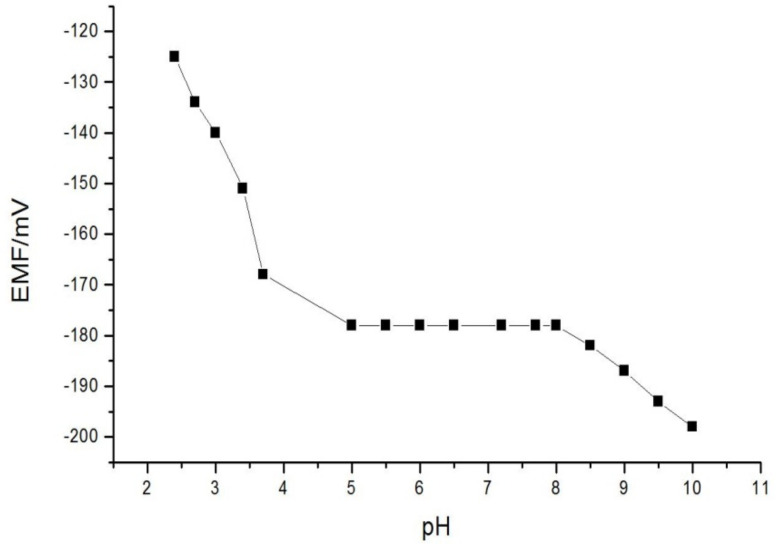
Influence of the developed electrode restraint on the pH in [10^−3^ M] aluminum ion concentration.

**Table 1 membranes-11-00504-t001:** Analytical validations of the developed Al^3+^ electrode matrix membrane.

Specific Slope/mV	58.00
Intercept/mV	−51.4 + 0.2
Limit of detection/mol dm^−3^	1.6 × 10^−7^
Measuring range/mol dm^−3^	2.5 × 10^−7^–1.5 × 10^−1^
Response time/s	12
Lifetime/d	150
pH range	5.0–8.0

**Table 2 membranes-11-00504-t002:** The values of selectivity coefficients (K) of Al^3+^ electrode.

Separate Solution Method (SSM)			Matched Potential Method
K	Ei = Ej	ai = aj	MPM
AlCl_3_	0.338 + 0.0320	0.366 + 0.010	0.366 + 0.0200
CdCl_2_	0.250 + 0.0060	0.333 + 0.020	0.295 + 0.0100
NiCl_2_	0.074 + 0.0030	0.162 + 0.004	0.278 + 0.0110
CoCl_2_	0.055 + 0.0120	0.077 + 0.001	0.006 + 0.0010
CaCl_2_	0.282 + 0.0070	0.363 + 0.050	0.312 + 0.0070
CuCl_2_	0.017 + 0.0002	0.078 + 0.003	0.014 + 0.0006

**Table 3 membranes-11-00504-t003:** Estimation of aluminum in foodstuff, real samples and standard alloys applying the proposed sensor.

Sample	Calibration Curve Method	Standard Addition Method
Sample Information mg/Kg	Al^3+^ Found mg/Kg	Relative Error %	V%	Sample Information mg/Kg	Al^3+^ Found mg/Kg	Relative Error %	V%
Bread	460	460.65	0.14	0.08	460	462.75	0.60	0.15
Flour	150	151.45	0.97	0.05	150	152.86	1.91	0.12
Rice	750	751.55	0.21	0.23	750	752.89	0.39	0.25
Tea-Leaves	550	551.86	0.34	0.12	550	552.95	0.54	0.16
Tomato Sauce	175	176.76	1.00	0.26	175	177.54	1.45	0.16
Chocolate	400	401.85	0.46	0.35	400	402.75	0.69	0.28
Granite	500	501.65	0.33	0.11	500	502.15	0.43	0.51
Basalt	500	501.75	0.35	0.13	500	502.65	0,53	0.26
Rhyolite	500	501.85	0.37	0.26	500	502.76	0.55	0.22
Copper-based alloy NBS164	250	251.95	0.78	0.37	250	252.75	1.10	0.23
Zinc-based alloy NIST94C	300	301.85	0.62	0.27	300	302.75	0.92	0.26

**Table 4 membranes-11-00504-t004:** Some interesting analytical validations of Al [DDTCT] in comparison with previously listed ion selective membrane sensors for aluminum determination.

Reference	Specific Slope (mV)	Linearity Concentration Range (M)	Duration Time	Detection Limit (M)	Working pH Range
This work Results	58.0	2.5 × 10^−7^–4.5 × 10^−1^	5 months	1.6 × 10^−7^	5.0–8.0
[2]	19.6 ± 0.4	1.0 × 10^−6^–1.0 × 10^−1^	>3 months	6.3 × 10^−7^	3.0–6.0
[7]	20 ± 0.2	1.6 × 10^−6^–1.0 × 10^−1^	3 months	6.0 × 10^−7^	3.0–8.5
[42]	19.3 ± 0.8	5.0 × 10^−6^–1.0 × 10^−2^	>2 months	2.5 × 10^−6^	3.5–5.0
[43]	19.5	1.0 × 10^−5^–1.0 × 10^−1^	1 month	3.2 × 10^−6^	2.25–3.25
[44]	29.5	1.0 × 10^−5^–1.0	2 months	1.0 × 10^−6^	-
[45]	18.5 ± 0.7	1.0 × 10^−6^–1.0 × 10^−2^	2 months	1.3 × 10^−7^	0.5–3.0
[46]	19.7 ± 0.1	3.2 × 10^−5^–1.0 × 10^−1^	2 months	3.2 × 10^−7^	3.5–5.0
[47]	19.8 ± 0.4	1.0 × 10^−6^–1.0 × 10^−1^	>4 months	4.6 × 10^−7^	2.0–6.0
[48]	19.0 ± 0.4	1.0 × 10^−6^–1.0 × 10^−1^	2 months	5.5 × 10^−7^	4.0–8.0
[49]	21.3 ± 0.18	7.0 × 10^−6^–1.0 × 10^−2^	-	6.0 × 10^−6^	-
[50]	19.6 ± 0.3	1.0 × 10^−7^–1.0 × 10^−2^	11 weeks	5.0 × 10^−8^	-

## Data Availability

The data that support the findings of this study are available on request from the corresponding author.

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
