# Peer review of "Selective Membrane Sensor for Aluminum Determination in Food Products, Real Samples and Standard Alloys"

_membranes, 2021, doi:10.3390/membranes11070504_

Round 1

Reviewer 1 Report

All my comments are addressed well. Now, this work can be accepted in its present form.

Author Response

                                                                  My Current address

                                                                   Dr. S. Khalil Mohamed            

                                                                   Food Nutrition Sci. Dept,

                                                                   Faculty of Science,                                                                    

                                                                   Taif University                                                                                                                                                          

                                                                   Taif 21944,

                                                                  P.O. Box 11099,

                                                                   Kingdom of Saudi Arabia

                                                                   Fax: + 96614915684

                                                                 Tel: 00966543372745

                         & [email protected]                                                                     

                                                                Thurs.,10June, 2021

Editor- in-Chief

        Membranes

Dear Editor,

           Thank you for your e-mail concerning the manuscript of ID reference : Membranes-1260722, entitled: “Selective Membrane Sensor for Aluminum Determination in Food Products, Real Samples and Standard Alloys ”                                                                                                                                    

            - With respect to the report of Reviewers 1, he stated that all his comments are  addressed well. Now , this work can be accepted in its present form.

            - With respect to the report of Reviewers 2, he recommended publication after some minor revisions.                     

            I have answered to Reviewer 2 . The reviewer comments were considered, the paper has been revised. The amendments and modifications done are in the following list:-

Answer to Reviewer No. 2

  1. The whole manuscript was re-checked with respect to grammar error and typos.
  2. Line 56 the phrase has been modified in the form of speech to be clear.
  3. Lines 59-63 becomes lines ( 59-70 ) after modification of the description to be clarify for a better comprehension of the reader.
  4. Line 71 becomes line 80: Saudia was corrected to Saudi.
  5. Line 77 become line 85 : dissolving was corrected to dissolved.
  6. Line 179 becomes line 188 : "Dependence of the Electrode Potential on the pH" is similar to the rest of the titles in the manuscript style.
  7. Lines 197-201 becomes lines 202-210 : The description was clarified after modification at the last revision by including more characterizations after adding some experimental results that support the claims at section 3.5 lifetime of the aluminum sensor.
  8. Line 252 becomes line 261:   As was corrected to as.
  9. Line 255 becomes line 264 : The was corrected to the.
  10. A marked copy of the revised manuscript is in the attached file in addition to the letter to the Editor.

         Thanking you in anticipation, please accept my best regards.

                                                                                      Yours Sincerely,

Prof. Dr. Sabry Khalil

Reviewer 2 Report

see attached file

Author Response

                                                                  My Current address

                                                                   Dr. S. Khalil Mohamed            

                                                                   Food Nutrition Sci. Dept,

                                                                   Faculty of Science,                                                                    

                                                                   Taif University                                                                                                                                                          

                                                                   Taif 21944,

                                                                  P.O. Box 11099,

                                                                   Kingdom of Saudi Arabia

                                                                   Fax: + 96614915684

                                                                 Tel: 00966543372745

                         & [email protected]                                                                     

                                                                Thurs.,10June, 2021

Editor- in-Chief

        Membranes

Dear Editor,

           Thank you for your e-mail concerning the manuscript of ID reference : Membranes-1260722, entitled: “Selective Membrane Sensor for Aluminum Determination in Food Products, Real Samples and Standard Alloys ”                                                                                                                                    

            - With respect to the report of Reviewers 1, he stated that all his comments are  addressed well. Now , this work can be accepted in its present form.

            - With respect to the report of Reviewers 2, he recommended publication after some minor revisions.                     

            I have answered to Reviewer 2 . The reviewer comments were considered, the paper has been revised. The amendments and modifications done are in the following list:-

Answer to Reviewer No. 2

  1. The whole manuscript was re-checked with respect to grammar error and typos.
  2. Line 56 the phrase has been modified in the form of speech to be clear.
  3. Lines 59-63 becomes lines ( 59-70 ) after modification of the description to be clarify for a better comprehension of the reader.
  4. Line 71 becomes line 80: Saudia was corrected to Saudi.
  5. Line 77 become line 85 : dissolving was corrected to dissolved.
  6. Line 179 becomes line 188 : "Dependence of the Electrode Potential on the pH" is similar to the rest of the titles in the manuscript style.
  7. Lines 197-201 becomes lines 202-210 : The description was clarified after modification at the last revision by including more characterizations after adding some experimental results that support the claims at section 3.5 lifetime of the aluminum sensor.
  8. Line 252 becomes line 261:   As was corrected to as.
  9. Line 255 becomes line 264 : The was corrected to the.
  10. A marked copy of the revised manuscript is in the attached file in addition to the letter to the Editor.

         Thanking you in anticipation, please accept my best regards.

                                                                                      Yours Sincerely,

Prof. Dr. Sabry Khalil

This manuscript is a resubmission of an earlier submission. The following is a list of the peer review reports and author responses from that submission.

Round 1

Reviewer 1 Report

This article introduced the use of DDTCT membrane modified ion-selective electrodes as a rapid detection assay for the detection aluminium ions. The impact of its pH value and its sensing performance were studied.

However, this article does not contain a necessary discussion of the mechanism of the specific detection of aluminium ions, which is of important for the readers. The related citation reference (such as ref-66) does not link me to the correct source of literature. In terms of ion-sensing performance, the authors provided just a few direct experimental results, such as the sensorgram or the calibration curves. Moreover, the original sensorgram data and calculation methods of sensitivity, LOD, and dynamic range should be clearly introduced in the manuscript. The only real-time sensorgram has given in Figure 2 also requires more repeats for further verification.

By considering these, I think this article still needs much more effort before it can be accepted by the journal.

Reviewer 2 Report

Khalil et al. have been showed interesting work on selective membrane-based sensor for the determination of the Aluminum content in the food products, real sample and standard alloys. Major changes need be address before the final decision.

1) Author has written abstract that mostly includes results section. Please revise the abstract. It should have subject background, short method, brief results and conclusive remark.

2) Please revise the referencing in the Introduction section. The introduction section itself 64 references. Please remove unnecessary references that too old and update the revise introduction section.

3) Please write a descriptive headings for the figure legends. Figure legends are very short and not much descriptive.

4) Figure 2 the response EMF vs t/s graph: Graph is the graph significance. Why the 10-3M you have mentioned in the heading you means that concentration is the mid point? Please elaborate on it. Also what is the what is the reason for a plateau for some the i.e. 10-6, 10-3, 10-2. Does the  

5)  What does the effect of pH 5-8 on the signal response. I am bit confused whether your checking effect of pH on signal response or vice a versa? Please clarify it in your figure 2 legends and also in description.  

6) What is the sensor system you are using? Please mention the description and diagram of it.

Reviewer 3 Report

see attached file

Reviewer 4 Report

The manuscript under review is devoted to the development of potentiometric Al3+ sensor. In my opinion this text represents severe violation of good practices in scientific publishing. The motivation for this opinion is as follows.

Introduction is poorly structured and does not put the work into the proper context. The scientific novelty is not clear. The authors should carefully analyze the literature available on Al(III) ISEs and highlight the drawbacks of already reported sensor membrane compositions. Citing 10 papers in one shot is hardly helpful for the reader. The motivation for using this particular substance as an ionophore is not clear. The authors state that DDCT “ has not only good sensitivity but also a very good selectivity coefficient.” The cited paper [66] DOI: 10.1039/AN9891400021 has no relation to the topic at all. The same paper is cited in the experimental section, but since it is devoted to the absolutely different topic, the nature of the active compound is not clear. The composition of the sensor membrane reported in section 2.7 contains two solvent-plasticizers. The motivation for this is not clear. The cited paper [63] has no relation to the study. And there are many other cases of inappropriate citations in the paper. The cited paper [65] reports on the development of selective Ga(III) electrode based on exactly the same ionophore as the one employed by the authors. The both of these cases simultaneously cannot be true, so either DDCT is selective to Ga(III), or to Al (III). Surprisingly, no studies to assess Ga(III) selectivity were performed.

The data in the Table 1, Table 2 and in Fig. 1 contradict each other. According to Fig.1 the sensor under study shows almost the same values of E0 for all studied sensors, and provides the slope around 20 mV/dec for all the ions. This means the absence of selectivity. At the same time Table 1 provides sensitivity of 58 mV/dec. While Table 2 provides some absolutely different values. With all due respect to the authors, after a careful reading all these numbers and figures look like unskillful fabrication. The text is full of inappropriate statements indicating severe lack of understanding of basic potentiometric concepts. English writing is very poor and requires thorough editing. At pH around 5 only about 50% of Al in aqueous solution will present in the form of Al3+ due to the hydrolysis, thus the indicated pH working range is nonsense.

I am deeply convinced that this study should not be accepted neither in Membranes, nor in any other journal.

Reviewer 5 Report

The work "Selective Membrane Sensor for Aluminum Determination in 2
Food Products, Real Samples and Standard Alloys" describes estimating the concentration of aluminum ions in food products, real samples, and standard alloy.

The abstract can be improved: it doesn't reflect the exact work of the manuscript.

The Introduction can be improved with some other related articles.

eg: Journal of The Electrochemical SocietyVolume 167Number 11

Journal of The Electrochemical SocietyVolume 167, Number 2

Ecotoxicology and Environmental Safety Volume 176, 30 July 2019, Pages 250-257.

The photograph of the prepared sensor can be added to the manuscript.

A few characterizations can be included in the manuscript.

The manuscript is lacking in characterizations.

The conclusion should be added with the future work.

The language of the manuscript must be polished.